# Is an Oral Health Status a Predictor of Functional Improvement in Ischemic Stroke Patients Undergoing Comprehensive Rehabilitation Treatment?

**DOI:** 10.3390/brainsci11030338

**Published:** 2021-03-07

**Authors:** Piotr Gerreth, Karolina Gerreth, Mateusz Maciejczyk, Anna Zalewska, Katarzyna Hojan

**Affiliations:** 1Private Dental Practice, 57 Kasztelanska Street, 60-316 Poznan, Poland; piotrger@hotmail.com; 2Postgraduate Studies in Scientific Research Methodology, Poznan University of Medical Sciences, 10 Fredry Street, 60-701 Poznan, Poland; 3Department of Risk Group Dentistry, Chair of Pediatric Dentistry, Poznan University of Medical Sciences, 70 Bukowska Street, 60-812 Poznan, Poland; karolinagerreth@poczta.onet.pl; 4Department of Hygiene, Epidemiology and Ergonomics, Medical University of Bialystok, 2C Adama Mic kiewicza Street, 15-022 Bialystok, Poland; mat.maciejczyk@gmail.com; 5Experimental Dentistry Laboratory, Medical University of Bialystok, 24A Marii Sklodowskiej-Curie Street, 15-276 Bialystok, Poland; azalewska426@gmail.com; 6Department of Occupational Therapy, Poznan University of Medical Sciences, 6 Swiecickiego Street, 60-781 Poznan, Poland; 7Department of Rehabilitation, Greater Poland Cancer Centre, 15 Garbary Street, 61-866 Poznan, Poland

**Keywords:** oral health, exercises, brain, teeth health, clinical assessment, neurology

## Abstract

The study’s aim was a clinical observation concerning the influence of oral health on functional status in stroke patients undergoing neurorehabilitation. This pilot cross-sectional clinical study was performed in 60 subacute phase stroke patients during 12 weeks of treatment. The program was patient-specific and consisted of neurodevelopmental treatment by a comprehensive rehabilitation team. The functional assessment was performed using the Barthel index (BI), Berg balance scale (BBS), functional independence measure (FIM), and Addenbrooke’s cognitive examination III (ACE III) scales. Oral health was assessed according to World Health Organization (WHO) criteria, and it was presented using DMFT, DMFS, gingival index (GI), and plaque index (PlI). Significant improvement in many functional scales was noticed. However, important differences in most dental parameters without relevant changes in GI and PlI after the study were not observed. Reverse interdependence (*p* < 0.05) was shown between physical functioning (BI, FIM, or BBS) with GI and PlI results, and most dental parameters correlated with ACE III. Using multivariate regression analysis, we showed that ACE III and BI are predictive variables for DMFT, just as FIM is for DS (*p* < 0.05). The present research revealed that poor oral health status in patients after stroke might be associated with inpatient rehabilitation results.

## 1. Introduction

Stroke has become the second leading cause of long-term disability and cognitive impairment. The disease may cause debilitating neurological deficiencies that result in motor, sensory, and cognitive deficits and deteriorated psychosocial functioning. Moreover, many researchers suggested an association between poor oral health, infections, and chronic systemic diseases such as ischemic stroke [1,2,3,4]. It is generally accepted that poor oral health may affect the general condition, whereas, on the other hand, systemic diseases might have oral manifestation and disturb the health of the oral cavity [5,6]. Such disorders as periodontal disease or untreated caries that often result in severe complications might intensify general diseases or even cause life-threatening situations [4,7]. Therefore, special attention in the dental care of such a group of patients is necessary. In recent meta-analyses, authors compared oral health between stroke patients and controls (healthy population). The results indicated that stroke patients had poorer oral health and worse periodontal status [8,9,10] as well as less frequent dental attendance behavior than controls [11,12]. Therefore, further research concerning oral health problems after stroke should be conducted, and effective management strategies need to be devised and implemented. It must be emphasized that only a few studies have evaluated oral health status according to stroke patients’ functional levels [13,14]. Tools are needed to measure progress in achieving the daily activity of rehabilitated stroke patients [15]. The quality of rehabilitation in stroke patients can be measured according to the level of independence achieved in daily activities. This goal seems to be of paramount importance in the rehabilitation of this group of patients. An adequate and rigorous assessment is vital for the rehabilitation physician (physiatrist) to establish proper physiotherapy and physical recovery outcomes, monitor progress in patients’ rehabilitation, and prescribe an adequate medicine treatment [16] with particular attention to polypharmacy and its possible consequences [17].

The dysfunctions associated with strokes, such as hemiparesis, visual field defects or cognitive impairment, can hinder independent oral care. Moreover, in disease-related oral health, stroke sufferers frequently experience numerous problems [1]. Therefore, many skills, e.g., manual dexterity, homonymous hemianopia, structural apraxia, and problems with the visual-spatial construction, need to be improved in such individuals. Additionally, multiple tooth loss is associated with cognitive impairment [18], and poor oral health can lead to severe complications, e.g., pneumonia, malnutrition, etc. [19], disturbing rehabilitation treatment progress in the early phases of stroke. Needless to say, intensive rehabilitation is necessary as soon as possible after a stroke incident to better improve motor, cognitive, and daily functioning of those groups of patients. Moreover, rehabilitation presents a positive effect on some predictive elements, such as the oxidative process [20,21]. However, medical professionals should consider that several factors can have an impact on recovery in hospitalized patients [22,23,24]. Despite establishing a relationship between oral health problems and several medical conditions, the need for cooperation between medical and dental professionals is little known regarding the impact of oral health problems on patients’ advances in rehabilitation treatment after stroke.

It must also be emphasized that knowledge concerning proper oral care in stroke patients that stay, e.g., at the rehabilitation centers and nursing home settings [25], is lacking. Some future research should focus on potential associations between diseases of the oral cavity and neglected oral hygiene [26]. Thus, more clinical studies are needed in the field to support medical professionals.

Therefore, the present study aimed to carry out a clinical observation concerning the influence of oral health on functional improvement in subacute stroke patients undergoing comprehensive neurorehabilitation.

## 2. Material and Methods

### 2.1. Study Design

This study was performed in the neurorehabilitation ward in Bonifraters’ Rehabilitation Center in Piaski-Marysin (Poland) between June 2019 and November 2019. It was a pilot cross-sectional clinical study and was approved by the Ethics Committee of the Poznan University of Medical Sciences (resolutions 59/19 and 890/19) in Poland. All participants were informed about the procedures and purpose of the research, and full written consent was obtained from them in accordance with the Declaration of Helsinki. The participation of all individuals in the research was voluntary. The patients were enrolled according to study criteria.

### 2.2. Study Criteria

The criteria for stroke sufferers to be included in the research were as follows: age of consent (>18 and <80 years old), consciousness and written and informed consent for oral and medical examination, good general condition, adequate capacity to follow the instructions during the examination, confirmed cerebral infarction based on brain computed tomography (CT) documentation and magnetic resonance imaging (MRI), and recovery from an acute phase of ischemic stroke in all brain areas. Neurological and CT findings were interpreted by a minimum of two independent and experienced specialists in neurology.

Patients with a medical history of pre-stroke dementia, hemorrhagic stroke, decreased consciousness, significant acute or chronic inflammatory factor, or neurological illness other than stroke (such as cerebral injury, tumor, sclerosis multiplex, Parkinson’s disease, etc.), and psychiatric disorders were excluded from the study. Other criteria that excluded from the research were unconsciousness, inability to provide informed consent for an oral examination, insufficient cooperation due to cognitive deficits, stroke recurrence during the subacute phase, heart failure resting oxygen saturation (SaO_2_) ≤94% or in NYHA (New York Heart Association) > II stage, lung disease (chronic obstructive pulmonary disease) or cardiovascular disease (angina or uncontrolled hypertension), or weight loss of >10% during the previous one month.

### 2.3. Assessment Schedule

In the group of patients, we used the following assessment schedule: Assessment I (baseline): during the first three days after admission to the neurorehabilitation ward. All patients were admitted to the neurorehabilitation unit in the subacute phase of stroke up to two weeks after being discharged from the stroke unit after the acute phase of stroke had resolved.

Assessment II: after twelve weeks of comprehensive inpatient rehabilitation treatment.

### 2.4. Rehabilitation Intervention

All patients were evaluated by a physiatrist or a neurorehabilitation specialist and then participated in an individually designed rehabilitation program according to their condition/level of disability. During rehabilitation treatment, every patient performed individual exercises 120 min a day, 6 days per week based on neuromuscular reeducation, coordination, balance, and gait training. Additionally, those patients had 1 h per day, 5 days a week of occupational, neuropsychological, and speech therapy. Moreover, the participants with dysphagia, dysarthria, or aphasia had individual sessions with speech therapists at the ward.

Nurses conducted twice-daily tooth brushing in the first weeks for all participants. Subsequently, with the improvement of hand function, the patients carried out supervised oral hygiene procedures. Occupational therapists delivered additional recommendations concerning self-brushing techniques.

Most participants followed the same diet divided into the baseline diet for most patients or diet for diabetes mellitus individuals. All the meals were prepared in the same hospital and distributed at the same time daily.

### 2.5. Measurements

#### 2.5.1. General Medical Information

Data on individuals’ general health status and condition were taken from patients’ documentation and included: sex, age, time since diagnosis of stroke, lesion of stroke, and additional medical and social history after admission to a rehabilitation ward. A physiatrist collected medical files during a semi-structured interview and based on medical records presented by the subject of the study.

#### 2.5.2. Oral Examination

Oral health status was assessed in a room separated from other patients in the rehabilitation ward. Following the World Health Organization criteria, a dental evaluation was performed in the artificial lighting of a headlamp [27]. Teeth were examined using a dental (ball-ended) probe and a plane mouth mirror. A patient was seated in a chair with his/her head resting against the wall with the dentist standing in front of them. Teeth were inspected wet, without previous professional cleaning. In some cases, the dentist used a cotton roll to remove debris from the tooth and gingiva’s surface. The tooth was considered as present within the oral cavity when any part of it was visible. Each surface (except the interproximal surface where there was no access) of all teeth was evaluated. Each tooth was evaluated and scored as sound, decayed (DT), extracted because of the carious process (MT), or filled due to caries (FT). The data obtained from the examination were used to calculate the DMFT index, which is the sum of DT, MT, and FT, and expresses dental caries experience. Dental caries prevalence was calculated as a percentage of subjects with a DMFT index above 0. The active carious lesion was recorded when the change showed a detectably softened area, undermined enamel, or unmistakable cavity. A probe was used for confirmation of visual evidence of caries. The tooth was considered missing (MT) when it had been removed due to caries complications (verified during history taking). Filled tooth (FT) was recorded when there was at least one permanent restoration placed to treat the carious process. Teeth with filling and carious cavity were assessed as those with caries (DT). The DMFS index was also evaluated. It is a sum of decayed surfaces of teeth (DS), surfaces of teeth missing due to the carious process (MS), and surfaces of teeth filled because of caries (FS). We adopted the generally accepted principle that there are four surfaces on anterior teeth (incisors and canines) and five surfaces on the posterior teeth (premolars and molars).

Separately, non-carious (v-shaped) cavities in the teeth’s cervical area were also evaluated in patients.

Gingival status was evaluated with the use of the gingival index (GI), according to Löe and Silness [28], and it was determined on teeth 16, 12, 24, 36, 32, and 44, with the following criteria: 0—absence of inflammation; 1—mild inflammation, a slight change in color and little change in texture; 2—moderate inflammation, moderate glazing, redness, edema, and hypertrophy, bleeding on pressure; 3—severe inflammation, marked redness and hypertrophy, tendency to spontaneous bleeding, ulceration [27]. The GI average scores were also grouped as follows: 0—no inflammation; 0.1–1.0—mild inflammation; 1.1–2.0—moderate inflammation; 2.1–3.0—severe inflammation.

Plaque index (PlI), according to Silness and Löe [29], was used to assess oral hygiene. It was determined on teeth 16, 12, 24, 36, 32, and 44, with the following criteria: 0—no plaque; 1—a film of plaque adhering to the free gingival margin and adjacent area of the tooth, where the plaque may be seen in situ only after application of disclosing solution or by using the probe on the tooth surface; 2—moderate accumulation of soft deposits within the gingival pocket or on the tooth and gingival margin that can be seen with the naked eye; 3—the abundance of soft matter within the gingival pocket and/or on the tooth and gingival margin [18,29]. The PlI scores were grouped as follows: 0—excellent; 0.1–1.0—good hygiene; 1.1–2.0—fair; 2.1–3.0—poor hygiene.

When the index tooth was missing, evaluation of the GI and PlI was performed on the neighboring one from the same group. If no tooth was present in a sextant qualifying for assessment, all the teeth present in that sextant were taken into account and examined, and the highest score was recorded as the sextant score.

The data of clinical examination were recorded in a dental chart specially designed for the research. They were used to calculate caries prevalence and severity, gingival index, and plaque index, as well as to establish the dental treatment needs of the patients.

The dental examination was performed by two dentists (P.G. and K.G.) after previous training and calibration by an experienced dental specialist (A.Z.) following the World Health Organization (WHO) criteria [14,27]. The intra-examiner and inter-examiner agreement for the DMFT was evaluated in another dental examination in 15 patients after two weeks, with a κappa value that was 0.96 and 0.94, respectively, whereas for GI and PlI, κ amounted to 0.95 and 0.92 as well as 0.96 and 0.93, respectively.

The time required for a dental examination and GI and PlI scores in each patient varied between 25–40 min. Following the WHO guideline, and due to unavailability of the equipment, dental radiographic examination for detecting approximal carious lesions was not carried out.

#### 2.5.3. Functional Assessment

##### Activities of Daily Living

To measure performance in activities of daily living (ADL), we used the Barthel index (BI), which is widely recognized as a metric for assessing a patient’s ability to perform certain activities when entering and leaving the hospital rehabilitation setting [30,31]. The purpose of its use is to determine the degree of a patient’s self-sufficiency. Patients answered the questions themselves if it was possible. In the case of speech or cognitive disorders, the responses were provided by a speech therapist or nurse. Specifically, tasks measure self-care (feeding, bathing, grooming, dressing, and toilet use), sphincter management (bladder and bowel management), bed/chair transfers, and locomotion (walking and stair use) [30,31]. A previous study reported by Ohura et al. [32] presented inter-rater reliability for the BI as 0.99 and intra-rater reliability for the value as 0.99 [32]. We measure our participants on the first day (admission day) and the last day of rehabilitation treatment in the ward.

##### Balance Assessment

We used a standard clinical test for the measurement of static and dynamic balance abilities, i.e., the Berg balance scale (BBS) [30,33]. The BBS is a qualitative measure that assesses balance via performing functional activities such as reaching, bending, transferring, and standing that incorporates most components of postural control: sitting and transferring safely between chairs; standing with feet apart, feet together, in single-leg stance, and feet in the tandem Romberg position with eyes opened or closed; reaching and stooping down to pick something off the floor [33]. Each item is scored along a 5-point scale, ranging from 0 to 4, each grade with well-established criteria. The total score of BBS ranges from 0 to 56 [33,34]. BBS was measured in our patients by physiotherapists during the first two days, and a control measure was taken on the last day at the ward. The Berg balance scale was assessed by Downs et al. [35]. The relative intra-rater reliability was 0.98, and the relative inter-rater reliability was 0.97 [35].

##### Assessment of Functional Independence

The functional independence measure (FIM) is a recommended scale consisting of an 18-item measurement tool that explores physical, psychological, and social function in stroke patients [36,37]. The FIM is a clinician-reported scale that assesses function in six areas, including self-care, continence, mobility, transfers, communication, and cognition [36]. Indeed, 13 items of the FIM represent a measure of motor function, and the other five items represent a measure of cognitive function [37]. The scale is used to assess how well a person can carry out basic ADL and thus how dependent the patient will be on others’ help. This tool is used to assess a patient’s level of disability and change in patient status in response to rehabilitation or medical intervention [36,37].

##### Cognitive Function Measurement

Cognitive assessment is recommended in stroke patients, but test completion may be complicated by stroke-related impairments [38]. Therefore, we used two scales to measure cognitive function in our participants: one that is a part of FIM as the cognitive questions part, and standardized tests to assess the participants’ global cognitive functioning, i.e., mini-mental state examination (MMSE) [39] and Addenbrooke’s cognitive examination III (ACE III) scale. We used ACE III as a screening technique that is capable of differentiating patients with and without cognitive impairment and composed of tests of attention, orientation, memory, language, visual perceptual, and visuospatial skills [40,41]. The score needs to be interpreted in the context of the patient’s overall medical history and examination [38,39,40,41]. Each participant underwent a cognitive assessment during a face-to-face examination by a clinical neuropsychologist in the first week (MMSE and ACE III) and at the last week of stay in the ward (ACE III).

### 2.6. Statistical Analysis

Statistical analysis was performed using GraphPad Prism 8.0 (GraphPad Software, Inc. La Jolla, USA) for MacOS. Data are presented as median with interquartile range (IQR) and mean (with standard deviation (SD) and 95% confidence intervals (CI)). The Shapiro–Wilk test was used to evaluate the distribution of results. Due to a lack of normal distribution, the Wilcoxon matched-pairs signed-rank test was used. Correlations between parameters were assessed using Spearman’s correlation coefficient. Multivariate analysis of the simultaneous impacts of many independent variables on one quantitative dependent variable was conducted by means of linear regression. Sex, age, smoking, number of strokes, ACE III, BI, FIM, and BBS were included as independent variables. A 95% CI was reported along with regression parameters. Statistical significance was assumed to be at *p* < 0.05.

The number of patients was determined a priori using the previous pilot study, assuming the test power of 0.9 (online sample size calculator, ClinCalc).

## 3. Results

### 3.1. Study Patients

Initially, 92 ischemic stroke sufferers were enrolled in the study, and finally, 60 participants who completed all assessments (medical, physical, and psychological evaluation) were qualified for the analysis (Figure 1).

A total of 60 patients (aged 63.4 ± 12.7) with moderate stroke severity NIHSS (National Institutes of Health Stroke Scale) scores 5.7 ± 3.7) who provided their consent to participate were included in statistical analysis. Basic information regarding the study participant characteristics (age, sex, types of disease, etc.) was collected on admission to the ward. The characteristics of the patients are presented in Table 1.

### 3.2. Analysis of Patient’s Oral Health Status during the Study Time

Table 2 shows dental examination results in the first three rehabilitation treatment days and after 12 weeks. We did not observe significant differences in most parameters, but we noticed relevant GI and PlI changes after study observation (statistically significant decrease in those parameters after 12 weeks of observation).

### 3.3. Analysis of Changes in Patient’s Functioning during the Study Time

After 12 weeks of inpatient rehabilitation treatment, we observed a great functional improvement. In Table 3, significant improvement in many functional scales (BI score, FIM scale, or BBS) and cognitive functions (in most ACE III subscales) was demonstrated.

### 3.4. Correlations between Functional Assessment and Oral Health Status

After correlations between functional assessment and oral health status results, statistically significant reverse interdependence between physical functional improvement (higher levels of BI, FIM scores, or BBS) with smaller GI and PlI results was noticed. Nevertheless, reverse correlation with ACE III total score was detected in most dental parameters (MT, FT, DMFT, MS, FS, and DMFS), but those dependencies were statistically significant only after 12 weeks of study time. Results of ACE III memory subscale correlated reverse with DMFT in both study times. In language function measurement, statistically significant interdependence with FT and FS after 12 weeks of rehabilitation (opposite to >0.05 after admission to the ward) was perceived. However, this function in ACE III observation had a reverse correlation with MT, DMFT, MS, and DMFS, and it was statistically significant only after the second study time measurement. All correlations assessed using Spearman’s correlation coefficient are presented in Table 4.

### 3.5. Multivariate Regression Analysis

Using multivariate regression analysis, we showed that DT, FT, DMFT, DS, MS, and DMFS depend on age, but we noticed that MT, GI, and PlI depend on the number of strokes in our study group. Interestingly, ACE III and BI were statistically significant predictive variables for DMFT, just as FIM for DS. Results of multifactorial regression analysis are shown in Table 5.

## 4. Discussion

Oral health and hygiene are known as crucial factors in stroke individuals because this group of patients has poorer clinical oral health across a range of many parameters, e.g., tooth loss, dental caries experience, or periodontal status [1,2,3,10,11].

In the present study, we observed oral health status as one of the possible predictors for functional improvement in subacute stroke patients who were subject to rehabilitation.

As previously confirmed, the oral function deteriorates quickly during the acute phase of stroke [42,43]. Very often, patients may demonstrate an impaired masticatory performance, possibly due to disturbed oral sensitivity and reduced tongue forces [44]. Additionally, facial asymmetry and the contra-lesion handgrip strength and tongue-palate contact during swallowing are impaired [44]. Moreover, dental attendance in patients after stroke is less frequent [11]. It is well known that daily oral hygiene procedures conducted by nurses or occupational therapists and self-care during in-hospital rehabilitation were effective in improving general oral health and facilitated plaque control in patients after stroke [45]. Despite international stroke guidelines that recommend routine oral hygiene care [14,19,46], it is often neglected during stroke rehabilitation by team members even though they have knowledge concerning the consequences of the stroke patient’s oral health conditions [12].

Therefore, we decided to carry out the observational cross-sectional study concerning the link between the functional changes in subacute stroke patients and their oral health status during 12 weeks of comprehensive rehabilitation treatment.

In our study, a wide range of functional measurements was used, including motor and cognitive assessment and dental examination at admission and discharge (after 12 weeks) of rehabilitation treatment. Cognitive impairment after stroke is a frequent consequence connected with neurological deficits such as sensory or motor impairment. In the current study, ACE III was applied to diagnose cognitive impairment and dementia, which correlated with many standardized neuropsychological tests [38,41,47,48]. Additionally, at admission, the neuropsychologist assessed the patients by MMSE as a screening stage of dementia. Dementia after stroke may encompass all types of cognitive dysfunctions [49]. A substantial improvement in cognitive parameters in all ACE III subscales after 12 weeks of rehabilitation treatment was observed.

Moreover, our patients’ oral hygiene and gingival status was not satisfactory at the beginning of observation (PlI = 1.9 and GI = 1.2) but was slightly improved at the end of a stay at the rehabilitation ward (1.6 and 0.99, respectively), and improvement occurred as overall health improved. The present research revealed high numbers of extracted teeth (MT = 18.0) as well as those with caries (DT = 2.6) or filled (FT = 2.4) in stroke patients. The worrying fact is the occurrence of multi-surface carious lesions in patients at the time of examinations (DS = 7.4) or those with fillings (FS = 5.8). The scores of dental caries intensity indexes (DMFT and DMFS) and their components (DT, DS, MT, MS, FT, and FS) had the same values during the first and second examination, indicating that there was no deterioration or improvement in dental health status. All patients had inpatient therapy and did not leave the health care center during 12 weeks of neurorehabilitation. On the other hand, there is no dental surgery at the institution where the treatment could be performed. Needless to say that patients with general illnesses, as indeed other individuals, should not have any focus on infection, such as periodontal disease or teeth with dental caries complications. It must be remembered that teeth provide a unique opportunity for bacterial invasion [7]. Untreated caries often result in necrosis and subsequently gangrene, and with time, infection of the periapical tissues occurs as a complication of such processes [7]. The lesion may progress into the neighboring soft tissues and cause life-threatening situations. Oral care plays a vital role in preventing dental caries and other oral disorders and maintaining the patients’ proper general condition and their recovery and activity [5,6]. Interestingly, it was proved that a systemic oral hygiene care program in stroke patients during their stay at the hospital was connected with a significant reduction in the risk of stroke-associated pneumonia that is a leading cause of mortality and disability related to this disorder [50]. Therefore, special attention in the dental care of such a group of patients is necessary. Indeed, during the dental examination at the center, the dentists had a unique opportunity to instruct the patients about oral hygiene. They recommended systematic, twice a day use of a soft toothbrush and normal toothpaste, after breakfast and in the evening, and rinsing of the mouth after lunch as it was recommended in the study of Murray and Scholten [19]. Mostly, the modified Bass method was applied to the individuals as it is easy to learn, the most effective, and the most widely accepted technique of toothbrushing [51]. This method was also used by other dentists in stroke patients [52]. Needless to say that proper hygienic requirements concerning dentures were also given. In numerous cases, there was a need for the assistance of a nurse or caregiver during the procedure, generally upon admission to the center, since the patients were not self-dependent due to problems with cognition, arm weakness, or aphasia. Interestingly, literature data point to the lack of staff training in oral hygiene techniques and oral assessments [53]. Therefore, it was indicated that training should be provided to nurses, nursing students, and health care assistants by qualified professionals such as dentists.

To the best of our knowledge, this is the first study that evaluated oral health conditions in relation to cognitive functions of stroke patients undergoing rehabilitation. The results showed a positive correlation between the numbers of teeth present in the oral cavity and cognitive functioning. Oral health examination revealed an adverse correlation between cognitive subscales such as memory with DMFT (*p* < 0.05 in both study times) and in language function (ACE subscales) with MT, MS, FT, FS, DMFT, and DMFS, especially after 12 weeks of rehabilitation treatment (*p* < 0.05). The highly multivariate nature of rehabilitation’s clinical study accompanied by a recovery trend for individual patients calls for advanced analytical approaches [54]. Therefore, multiple regression is a common way to assess co-variations between and among different variables [54,55]. It can be used to consider multiple independent variables when calculating a complex data set. We noticed ACE III and BI were statistically significant predictive variables for DMFT, just as FIM was for DS. Other researchers carrying out the studies in stroke patients did a similar observation; however, without considering the aspect of comprehensive neurorehabilitation [19,56,57]. Poor dental health is associated with nutritional deficiency and chronic inflammation [11,13] and connected with cognitive impairment [56,57]. The present observational study confirmed a positive aspect of comprehensive rehabilitation, including dental and nutrition care, oral health, cognitive and functional function, and interdependence. In a previous study, Shiraishi et al. [13] showed that poor oral health status was associated with the worse activity of daily living after several weeks of post-acute in-hospital rehabilitation. In the present study, we had a similar general conclusion that “impaired oral health status may be associated with rehabilitation outcomes in hospitalized patients.” However, those authors [13] used the revised oral assessment guide (ROAG) to measure oral health at admission to the rehabilitation ward and only motor scales in FIM at discharge. Thus, it could be challenging to compare the results.

In the present research, patients even with single teeth were taken into consideration bearing in mind the fact that regardless of the number of teeth present in the oral cavity, it is necessary to take care of oral hygiene properly. On the other hand, a high DMFT index shows previous dental problems, i.e., dental caries and their complications, which might have deleterious effects on the organism and affect the general health of the individual.

Analysis of the results concerning the physical functioning of our study participants showed a positive aspect of rehabilitation on patients’ independence. This is in accordance with the results of numerous researchers [15,58,59,60]. However, we confirmed more improvement in physical function (BI, BBS, and FIM) with adverse (*p* < 0.05 correlation) oral health status in GI and PlI. This is a crucial observation that was not described in detail in the literature previously. Only Suzuki et al. [61] found that post-stroke rehabilitation treatment with dental hygienist care improved FIM score and discharged ward time results. Therefore, we could emphasize that the better oral health in stroke patients is, the more effective neurorehabilitation results. On the other hand, comprehensive rehabilitation is essential since it ensures improvement in individuals’ manual dexterity and increases oral hygiene and the gingiva’s better condition.

Indeed, the present research has some limitations that require discussion. Firstly, we have not included individuals that have suffered cerebrovascular accidents or patients after a stroke that were not undergoing intensive neurorehabilitation therapy. It would be interesting to make observations and to compare the progression between these groups during the same time of observations. Secondly, radiographs of the patients’ teeth for diagnosis of interproximal caries were not taken. Consequently, some carious cavities might be underestimated.

This study’s main strengths are that it is the first research where authors assessed oral health and functional assessments in physical mobility and cognitive function in subacute stroke patients who had comprehensive neurorehabilitation. The second was a medical examination by a multi-specialist medical team: dentist (oral assessment), physiatrist (physical assessment), and neuropsychologist (cognitive assessment) using popular and commonly used clinical tests based on numerous scientific studies. Moreover, our cross-sectional study limited the ability to draw a valid conclusion, such as the relationship between oral health and cognition or motor tools. However, further studies should determine the biological mechanisms and establish an intervention strategy to optimize oral health status during stroke patients’ rehabilitation treatment.

This pilot study has some other strengths. Firstly, we considered only patients after ischemic stroke because, in global terms, this type of stroke is almost four-fold more frequent than hemorrhagic ones [15]. Therefore, the study group was homogeneous. The neurological status of all patients included in the study was confirmed by early brain imaging using CT or MRI in the stroke unit. Moreover, few clinically useful tests, i.e., BI, BBS, and FIM, widely used in international classification to measure disability and functioning, were applied [26,32,33,34,62,63]. Therefore, the disability that affects daily living activities and the potential functional improvement of patients undergoing a rehabilitation program could be assessed. Those scales may help to develop precision medicine tools and evaluate a rehabilitative path in patients after stroke [15,63].

In summary, one may say that as in other patients with general illnesses, in stroke individuals, very close interdisciplinary collaboration between different medical specialists, e.g., neurologists, neurorehabilitation physicians, physiatrists, psychologists, occupational therapists, and dentists, is recommended. There is a peculiar role of oral health medical staff and other practitioners in the therapy of such individuals and preventing and counseling social diseases, including oral ones [64].

Therefore, it must be particularly underlined that keeping good oral hygiene in stroke patients affected by paresis of various degrees might be extremely challenging. Thus, manual dexterity is vital for toothbrushing, which is essential for maintaining proper oral hygiene and health [64]. However, to achieve this, intensive rehabilitation after a stroke incident is necessary.

## 5. Conclusions

Poor oral health status in individuals after a stroke incident might be associated with rehabilitation results in hospitalized patients. Therefore, stroke patients should be systematically examined and supervised by dental professionals who need to cooperate with other medical staff to treat those individuals to provide them with comprehensive evaluation and care.

## Figures and Tables

**Figure 1 brainsci-11-00338-f001:**
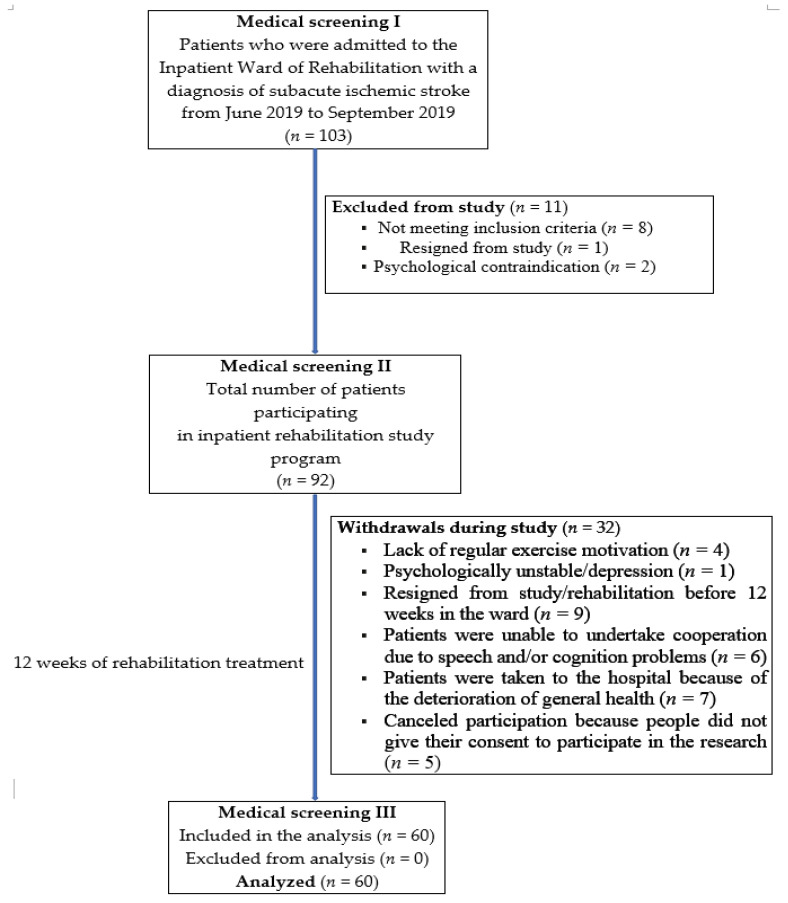
The flow of participants throughout the study.

**Table 1 brainsci-11-00338-t001:** Characteristics of stroke-study patients.

Patients Demographic and Clinical Characteristics		Study Group N (%)
Sex	Male	34 (56.66)
	Female	26 (43.33)
Formal education	Primary	8 (13.33)
	Vocational	17 (28.33)
	Secondary	19 (31.66)
	University	16 (26.66)
Location of daily life	Urban center (over 150.000 inhabitants)	27 (45.00)
	Small town (10.000–150.000 inhabitants)	17 (28.33)
	Rural area or small village (less than 10.000 inhabitants)	16 (26.66)
Housing status	Living home with family member	41 (68.66)
	Living alone	19 (31.66)
Brain lesion side	Left	27 (45.00)
	Right	33 (55.00)
Comorbidities	BMI >30	28 (46.66)
	Hypertension	42 (70.00)
	Diabetes	28 (46.66)
	Epilepsy	6 (10.00)
	Arteriosclerosis	37 (61.66)
	Limb thrombosis in the past	4 (6.66)
	Glucose intolerance	10 (16.66)
	Myocardial infarction	6 (10.00)
	Atrial fibrillation	18 (30.00)
	Angina pectoris	4 (6.66)
	Chronic renal failure	6 (10.00)
	Chronic obstructive pulmonary disease	3 (5.00)
	Hyperlipidemia	39 (65.00)
	Bronchial asthma	4 (6.66)

**Table 2 brainsci-11-00338-t002:** Analysis of patient’s oral health status during the study time.

Parameter	Assessment I		Assessment II		*p*-Value
	Median (IQR)	Mean (SD) (95% CI)	Median (IQR)	Mean (SD) (95% CI)	
Number of teeth present in oral cavity	14 (2–24)	14 (11) (10–18)	14 (2–24)	14(11) (10–18)	1.0
DT	1 (0–4)	2.6 (3.6) (1.7–3.5)	1 (0–4)	2.6 (3.6) (1.7–3.5)	1.0
MT	20 (8–28)	18 (10) (15–20)	20 (8–28)	18 (10) (15–20)	1.0
FT	0 (0–3)	2.4 (4.5) (1.2–3.6)	0 (0–3)	2.4 (4.5) (1.2–3.6)	1.0
DMFT	25 (17–28)	23 (7) (21–25)	25 (17–28)	23 (7) (21–25)	1.0
DS	2 (0–9)	7.4 (12) (4.3–10)	2 (0–9)	7.4 (12) (4.3–10)	1.0
MS	92 (40–147)	86 (51) (73–100)	92 (40–147)	86 (51) (73–100)	1.0
FS	0 (0–5.5)	5.8 (13) (2.4–9.1)	0 (0–5.5)	5.8 (13) (2.4–9.1)	1.0
DMFS	113 (57–147)	100 (43) (88–111)	113 (57–147)	100 (43) (88–111)	1.0
GI	1.2 (0.5–2)	1.2 (0.87) (0.98–1.5)	0.83 (0–1.8)	0.99 (0.9) (0.71–1.3)	<0.001
PlI	2 (1–3)	1.9 (0.93) (1.6–2.2)	1.5 (0.5–2.5)	1.6 (1) (1.2–1.9)	<0.001

IQR—interquartile range; SD—standard deviation; CI—confidence intervals; DMFT—caries severity index, that is a sum of decayed teeth (DT), teeth missing due to carious process (MT), and teeth filled because of caries (FT); DMFS—a sum of decayed surfaces of tooth (DS), surfaces of tooth missing due to carious process (MS), and surfaces of tooth filled because of caries (FS); GI—gingival index; PlI—plaque index.

**Table 3 brainsci-11-00338-t003:** Analysis of changes in patient’s functioning during the study time.

Parameter	Assessment I		Assessment II		*p*-Value
	Median (IQR)	Mean (SD) (95% CI)	Median (IQR)	Mean (SD) (95% CI)	
BI	12 (7–13)	11 (3.9) (9.5–12)	19 (15–20)	16 (5.1) (15–18)	<0.001
FIM	89 (60–109)	82 (32) (74–80)	115 (96–126)	104 (31) (96–112)	<0.001
BBS	36 (17–42)	30 (16) (26–31)	50 (37–54)	43 (15) (39–47)	<0.001
ACE III total scale	72 (57–82)	66 (23) (60–72)	81 (68–90)	76 (22) (70–82)	<0.001
ACE memory	18 (13–22)	17 (6.8) (15–18)	21 (18–24)	19 (6.3) (18–21)	<0.001
ACE attention and orientation	14 (12–17)	13 (4.3) (12–15)	16 (14–18)	15 (3.5) (14–16)	<0.001
ACE verbal fluency	8 (5–11)	7.5 (3.9) (6.5–8.5)	10 (7–12)	9.1 (3.6) (8.1–10)	<0.001
ACE language function	21 (18–24)	19 (7.3) (17–21)	24 (20–25)	21 (6.1) (20–23)	<0.001
ACE visual-spatial functions	11 (8–16)	11 (4.5) (9.7–12)	13 (11–15)	13 (3.6) (12–13)	<0.001

IQR—interquartile range; SD—standard deviation; CI—confidence intervals; BI—Barthel index; FIM—functional independence measure; BBS—Berg balance score; ACE—Addenbrooke’s cognitive examination III.

**Table 4 brainsci-11-00338-t004:** Correlations between functional assessment and oral health status.

	Study Time	Number of Teeth Present in Oral Cavity	DT	MT	FT	DMFT	DS	MS	FS	DMFS	GI 1	GI 2	PlI 1	PlI 2
BI	I	−0.203	−0.118	0.013	0.019	−0.071	−0.14	0.021	0.019	−0.043	−0.4 **	−0.42 **	−0.363 *	−0.344 *
	II	0.281	0.048	−0.208	0.238	−0.248	0.019	−0.203	0.231	−0.212	−0.293	−0.232	−0.338 *	−0.215
FIM	I	−0.001	−0.134	−0.078	0.066	−0.2	−0.166	−0.062	0.069	−0.134	−0.394 **	−0.433 **	−0.408 **	−0.399 **
	II	0.126	−0.087	−0.137	0.155	−0.223	−0.114	−0.121	0.15	−0.184	−0.403 **	−0.407 **	−0.424 **	−0.381 *
BBS	I	0.018	−0.124	−0.082	0.038	−0.195	−0.138	−0.066	0.045	−0.122	−0.33 *	−0.423 **	−0.363 *	−0.4 **
	II	0.217	−0.045	−0.218	0.182	−0.299	−0.082	−0.199	0.183	−0.238	−0.389 **	−0.386 *	−0.465 **	−0.407 **
ACE III	I	0.179	0.006	−0.19	0.161	−0.286 *	−0.023	−0.183	0.156	−0.212	−0.154	−0.121	−0.211	−0.117
	II	0.395 *	−0.03	−0.312 *	0.346 **	−0.339 **	−0.054	−0.325*	0.336 **	−0.329*	−0.257	−0.189	−0.319*	−0.208
ACE—attention and orientation	I	0.084	0.002	−0.11	0.09	−0.204	−0.046	−0.107	0.092	−0.15	−0.244	−0.221	−0.3*	−0.243
	II	0.143	−0.061	−0.18	0.242	−0.221	−0.109	−0.223	0.233	−0.245	−0.16	−0.121	−0.293	−0.236
ACE—memory	I	0.349	−0.03	−0.233	0.16	−0.28*	−0.057	−0.215	0.149	−0.23	−0.069	0.027	−0.1	0.045
	II	0.309	−0.03	−0.235	0.222	−0.273*	−0.062	−0.234	0.211	−0.237	−0.023	0.043	−0.113	0.023
ACE—verbal fluency	I	0.102	0.006	−0.094	0.035	−0.123	−0.02	−0.079	0.034	−0.088	−0.079	−0.085	−0.174	−0.134
	II	0.333	0.067	−0.244	0.225	−0.221	0.046	−0.221	0.225	−0.206	−0.175	−0.18	−0.28	−0.224
ACE—language function	I	0.24	−0.085	−0.161	0.191	−0.202	−0.129	−0.158	0.192	−0.17	−0.185	−0.187	−0.239	−0.197
	II	0.411	−0.113	−0.274 *	0.35 **	−0.344 **	−0.142	−0.298*	0.352 **	−0.322*	−0.22	−0.193	−0.275	−0.224
ACE—visual−spatial function	I	−0.155	−0.068	−0.121	0.18	−0.18	−0.132	−0.138	0.187	−0.182	−0.203	−0.194	−0.235	−0.174
	II	0.207	−0.051	−0.241	0.311	−0.209	−0.105	−0.256	0.318*	−0.227	−0.225	−0.224	−0.238	−0.192

DMFT—caries severity index, that is a sum of decayed teeth (DT), teeth missing due to carious process (MT), and teeth filled because of caries (FT); DMFS—a sum of decayed surfaces of tooth (DS), surfaces of tooth missing due to carious process (MS), and surfaces of tooth filled because of caries (FS); GI—gingival index; PlI—plaque index; BI—Barthel index; FIM—functional independence measure; BBS—Berg balance score; ACE—Addenbrooke’s cognitive examination III; * *p* < 0.05; ** *p* < 0.01.

**Table 5 brainsci-11-00338-t005:** Multifactorial regression analysis of the assessed parameters.

Dependent Variable	Independent Variable								
		Age	Sex	Tobacco Smoking	Number of Strokes	ACE III	BI	FIM	BBS
Number of teeth present in oral cavity	EE	−0.266	−1.545	1	−1.021	0.0594	−0.957	0.04787	0.1623
	95% CI	−0.5506 to 0.01865	−7.676 to 4.586	−6.576 to 8.576	−3.678 to 1.636	−0.1215 to 0.2403	−1.980 to 0.06632	−0.2245 to 0.3203	−0.2490 to 0.5736
	*p*-value	0.0664	0.6156	0.7924	0.4445	0.5133	0.0662	0.726	0.4324
DT	EE	−0.1042	−0.2103	0.2715	−0.3998	0.008942	0.03388	−0.03951	0.04047
	95% CI	−0.1714 to −0.03701	−1.578 to 1.158	−1.320 to 1.863	−1.169 to 0.3694	−0.02735 to 0.04523	−0.2210 to 0.2888	−0.1020 to 0.02296	−0.05794 to 0.1389
	*p*-value	0.0027 **	0.7611	0.7358	0.3051	0.6262	0.7926	0.2126	0.4167
MT	EE	0.004214	0.319	0.118	0.279	−0.004451	0.0358	−0.0002452	−0.02673
	95% CI	−0.01347 to 0.02191	−0.06859 to 0.7065	−0.3337 to 0.5697	0.08872 to 0.4692	−0.01607 to 0.007167	−0.03703 to 0.1086	−0.01714 to 0.01665	−0.05525 to 0.001799
	*p*-value	0.6367	0.1053	0.6046	0.0046 **	0.4479	0.3307	0.977	0.0659
FT	EE	−0.1083	−0.6938	−1.465	−0.5722	0.02142	−0.0005962	−0.03311	0.06244
	95% CI	−0.1886 to −0.0280	−2.329 to 0.9411	−3.367 to 0.4371	−1.492 to 0.3471	−0.02195 to 0.06479	−0.3040 to 0.3052	−0.1078 to 0.04154	−0.05516 to 0.1801
	*p*-value	0.0087 **	0.402	0.1297	0.2199	0.3297	0.9969	0.3811	0.2949
DMFT	EE	0.1841	−1.815	−1.513	−0.5752	−0.06282	0.4849	−0.0427	−0.1218
	95% CI	0.0593 to 0.3089	−4.359 to 0.7212	−4.469 to 1.442	−2.004 to 0.8533	−0.1302 to 0.00457	0.01156 to 0.9582	−0.1587 to 0.07329	−0.3045 to 0.06098
	*p*-value	0.0042 **	0.1586	0.3124	0.4264	0.0497 *	0.0448 *	0.467	0.1893
DS	EE	−0.3086	−0.1191	1.227	−1.054	0.02	0.1998	−0.2207	0.2313
	95% CI	−0.5359 to −0.08117	−4.748 to 4.510	−4.159 to 6.612	−3.657 to 1.549	−0.1028 to 0.1427	−0.6627 to 1.1062	−0.4321 to −0.009355	−0.1017 to 0.5643
	*p*-value	0.0083 **	0.9594	0.6525	0.4238	0.7474	0.647	0.0409 *	0.1714
MS	EE	1.998	−6.178	−3.742	−1.757	−0.425	1.931	0.1456	−1.031
	95% CI	1.133 to 2.863	−23.75 to 11.48	−24.24 to 16.75	−11.66 to 8.148	−0.8928 to 0.04226	−1.351 to 5.213	−0.6587 to 0.9498	−2.298 to 0.2367
	*p*-value	<0.0001 **	0.4883	0.718	0.7258	0.0742	0.245	0.7204	0.1097
FS	EE	−0.1404	−0.8427	−3.474	−1.908	0.05655	−0.02189	−0.1204	0.2435
	95% CI	−0.3936 to 0.1128	−5.997 to 4.312	−9.470 to 2.523	−4.806 to 0.9901	−0.08018 to 0.1933	−0.9384 to 0.8822	−0.3558 to 0.1149	−0.1273 to 0.6142
	*p*-value	0.2741	0.7464	0.2533	0.1946	0.4141	0.964	0.3126	0.1958
DMFS	EE	1.549	−7.14	−5.99	−4.719	−0.3485	2.152	−0.1956	−0.5561
	95% CI	0.8175 to 2.280	−22.03 to 7.749	−23.310 to 11.33	−13.09 to 3.654	−0.7435 to 0.04654	−0.6218 to 4.927	−0.8755 to 0.4843	−1.627 to 0.5150
	*p*-value	<0.0001 **	0.3439	0.4945	0.2663	0.0832	0.127	0.5696	0.3056
PlI	EE	−0.00303	0.4128	0.2507	0.305	−0.006446	0.01106	−0.000569	−0.02626
	95% CI	−0.0216 to 0.0156	0.00127 to 0.8244	−0.2395 to 0.7408	0.1037 to 0.5063	−0.01923 to 0.00634	−0.0656 to 0.08773	−0.01759 to 0.01873	−0.05651 to 0.00398
	*p*-value	0.7469	0.0493 *	0.3115	0.0035 **	0.3185	0.7745	0.9504	0.0878
GI	EE	−0.01068	0.1511	−0.04456	0.2454	−0.000729	−0.004064	−0.003209	−0.01496
	95% CI	−0.02783 to 0.006461	−0.2277 to 0.5299	−0.4957 to 0.4066	0.06012 to 0.4306	−0.0125 to 0.01104	−0.07462 to 0.0665	−0.01993 to 0.01351	−0.0428 to 0.01288
	*p*-value	0.186	0.4541	0.7842	0.0101 *	0.9021	0.909	0.7032	0.288

EE—estimate; CI—confidence intervals; DMFT—caries severity index, that is a sum of decayed teeth (DT), teeth missing due to carious process (MT), and teeth filled because of caries (FT); DMFS—a sum of decayed surfaces of tooth (DS), surfaces of tooth missing due to carious process (MS), and surfaces of tooth filled because of caries (FS); GI—gingival index; PlI—plaque index; BI—Barthel index; FIM—functional independence measure; BBS—Berg balance score; ACE—Addenbrooke’s cognitive examination III total score; * *p* < 0.05; ** *p* < 0.01.

## Data Availability

The datasets generated for this study are available on request to the corresponding author.

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
