# Peer review of "Is an Oral Health Status a Predictor of Functional Improvement in Ischemic Stroke Patients Undergoing Comprehensive Rehabilitation Treatment?"

_brainsci, 2021, doi:10.3390/brainsci11030338_

Round 1

Reviewer 1 Report

Authors presented a very interesting study about the relationship between oral health status and functional improvement in ischemic stroke patients undergoing neurorehabilitation treatment.

It is an interesting well presented research but some concepts should be added. 

In Introduction: "prescribe an adequate medicine treatment [11] with particular attention to the polypharmacy and its possible consequence ["Association of Polypharmacy With 1-Year Trajectories of Cognitive and Physical Function in Nursing Home Residents: Results From a Multicenter European Study" Vetrano 2018]
Please consider that several factors could impact on recovery in hospitalized patients  [Padua et al."Cognitive reserve as a useful variable to address robotic or conventional upper limb rehabilitation treatment after stroke"; Vetrano et al. 2018 "Health determinants and survival in nursing home residents in Europe: Results from the SHELTER study"; "RANK/RANKL/OPG pathway: genetic association with history of ischemic stroke in Italian population" Biscetti et al. 2016] 

It is important to say that Rehabilitation presents a positive effect on some predictive elements such as oxidative process ["Myeloperoxidase levels and mortality in frail community-living elderly individuals"Giovannini S et al. 2010 - and "Selenium Concentrations and Mortality Among Community-Dwelling Older Adults: Results from ilSIRENTE Study" Giovannini 2018].

Author Response

Concern of the Reviewer: Moderate English changes required.

Authors’ response: We have checked the manuscript once again to correct mistakes. The manuscript has been proofread by professional sworn translator Ms. Anna Adamkiewicz(licence no. [email protected]).

Concern of the Reviewer: Authors presented a very interesting study about the relationship between oral health status and functional improvement in ischemic stroke patients undergoing neurorehabilitation treatment.

It is an interesting well presented research but some concepts should be added. 

In Introduction: "prescribe an adequate medicine treatment [11] with particular attention to the polypharmacy and its possible consequence ["Association of Polypharmacy With 1-Year Trajectories of Cognitive and Physical Function in Nursing Home Residents: Results From a Multicenter European Study" Vetrano 2018]
Please consider that several factors could impact on recovery in hospitalized patients  [Padua et al."Cognitive reserve as a useful variable to address robotic or conventional upper limb rehabilitation treatment after stroke"; Vetrano et al. 2018 "Health determinants and survival in nursing home residents in Europe: Results from the SHELTER study"; "RANK/RANKL/OPG pathway: genetic association with history of ischemic stroke in Italian population" Biscetti et al. 2016] 

It is important to say that Rehabilitation presents a positive effect on some predictive elements such as oxidative process ["Myeloperoxidase levels and mortality in frail community-living elderly individuals" Giovannini S et al. 2010 - and "Selenium Concentrations and Mortality Among Community-Dwelling Older Adults: Results from ilSIRENTE Study" Giovannini 2018].

Authors’ response: Thank you very much for your suggestion. We have added information to the introduction with the references that have been included in the list.

“In recent meta-analyses authors compared oral health between stroke patients and controls (healthy population). Results indicated, that stroke patients had poorer oral health and worse periodontal status [8-10] as well as less frequent dental attendance behaviour than controls [11,12]. Therefore, further research concerning oral health problems after stroke should be conducted and effective management strategies need to be devised and implemented. It must be emphasized that only a few studies have evaluated oral health status according to stroke patients' functional level [13,14]. Tools are needed to measure progress in achieving daily activity of rehabilitated stroke patients. [15]. The quality of rehabilitation in stroke patients can be measured according to the level of independence achieved in daily activities. This goal seems to be of paramount importance in the rehabilitation of this group of patients. An adequate and rigorous assessment is vital for the rehabilitation physician (physiatrist) to establish proper physiotherapy and physical recovery outcomes, monitor progress in patients’ rehabilitation, and prescribe an adequate medicine treatment [16] with particular attention to the polypharmacy and its possible consequence [17].”

Lafon, A.; Pereira, B.; Dufour, T.; Rigouby, V.; Giroud, M.; Béjot, Y.; Tubert-Jeannin, S. Peridontal disease and stroke: a met-analysis of cohort studies. Eur J Neurol. 2014, 21, 1155-61, e66-7. doi: 10.1111/ene.12415.

Fagundes, N.C.F.; Almeida, A.P.C.P.S.C.; Vilhena, K.F.B.; Magno, M.B.; Maia, L.C.; Lima, R.R. Periodontitis as a risk factor for stroke: a systematic review and meta-analysis. Vasc Health Risk Manag. 2019, 15, 519-532. doi: 10.2147/VHRM.S204097.

Zeng, L.N.; Rao, W.W.; Luo, S.H.; Zhang, Q.E.; Hall, B.J.; Ungvari, G.S.; Chen, L.G.; Xiang, Y.T. Oral health in patients with stroke: a meta-analysis of comparative studies. Top Stroke Rehabil. 2020, 27, 75‐80, doi:10.1080/10749357.2019.1656413.

Dai, R.; Lam, O.L.; Lo, E.C.; Li, L.S.; Wen, Y.; McGrath, C. A systematic review and meta-analysis of clinical, microbiological, and behavioural aspects of oral health among patients with stroke. J Dent. 2015, 43, 171-80, doi: 10.1016/j.jdent.2014.06.005.

Ajwani, S.; Jayanti, S.; Burkolter, N.; Anderson, C.; Bhole, S.; Itaoui, R.; George, A. Integrated oral health care for stroke patients - a scoping review. J Clin Nurs. 2017, 26, 891-901, doi: 10.1111/jocn.13520.

Roman, N.; Miclaus, R.; Repanovici, A.; Nicolau, C. Equal Opportunities for Stroke Survivors’ Rehabilitation: A Study on the Validity of the Upper Extremity Fugl-Meyer Assessment Scale Translated and Adapted into Romanian. Medicina 2020, 56, 409, doi:10.3390/medicina56080409.

Vetrano, D.L.; Villani, E.R.; Grande, G.; Giovannini, S.; Cipriani, M.C.; Gravina, E.M.; Bernabei, R.; Onder, G. Association of Polypharmacy With 1-Year Trajectories of Cognitive and Physical Function in Nursing Home Residents: Results From a Multicenter European Study. J Am Med Dir Assoc 2018, 19, 710-713, doi: 10.1016/j.jamda.2018.04.008.

“Needless to say, intensive rehabilitation is necessary as soon as possible after stroke incident to better improve motor, cognitive, and daily functioning of those group of patients. Moreover, rehabilitation presents a positive effect on some predictive elements such as oxidative process [20,21]. However, medical professionals should consider that several factors can impact on recovery in hospitalized patients [22-24]. Despite establishing a relationship between oral health problems and several medical conditions, the need for cooperation between medical and dental professionals is little known regarding the impact of oral health problems on patients' advances in rehabilitation treatment after stroke.

It must be also emphasized that knowledge concerning proper oral care in stroke patients that stay, e.g, at the rehabilitation centers and nursing home settings [25] is lacking. Some future research should focus on potential associations between diseases of oral cavity and neglected oral hygiene [26]. Thus, more clinical studies are needed in the field to support medical professionals.”

Giovannini, S.; Onder, G.; Leeuwenburgh, C.; Carter, Ch.; Marzetti, E.; Russo, A.; Capoluongo, E.; Pahor, M.; Bernabei, R.; Landi, F. Myeloperoxidase levels and mortality in frail community-living elderly individuals. J Gerontol A Biol Sci Med Sci 2010, 65, 369-76, doi: 10.1093/gerona/glp183.

Giovannini, S.; Onder, G.; Lattanzio, F.; Bustacchini, S.; Di Stefano, G.; Moresi, R.; Russo, A.; Bernabei, R.; Landi, F. Selenium Concentrations and Mortality among Community-Dwelling Older Adults: Results from IlSIRENTE Study. J Nutr Health Aging 2018, 22, 608-612. doi: 10.1007/s12603-018-1021-9.

Padua, I.; Imbimbo, I.; Aprile, I.; Loreti, C.; Germanotta, M.; Coraci, D.; Piccinini, G.; Pazzaglia, C.; Santilli, C.; Cruciani, A.; Carrozza, M.C.; FDG Robotic Rehabilitation Group†. Cognitive reserve as a useful variable to address robotic or conventional upper limb rehabilitation treatment after stroke: a multicentre study of the Fondazione Don Carlo Gnocchi. Eur J Neurol 2020, 27, 392-398, doi: 10.1111/ene.14090.

Vetrano, D.L.; Collamati, A.; Magnavita, N.; Sowa, A.; Topinkova, E.; Finne-Soveri, H.; van der Roest, H.G.; Tobiasz-Adamczyk, B.; Giovannini, S.; Ricciardi, W.; Bernabei, R.; Onder, G.; Poscia, A. Health determinants and survival in nursing home residents in Europe: Results from the SHELTER study. Maturitas. 2018, 107, 19-25. doi: 10.1016/j.maturitas.2017.09.014.

Biscetti, F.; Giovannini, S.; Straface, G.; Bertucci, F.; Angelini, F.; Porreca, C.; Landolfi, R.; Flex, A.  RANK/RANKL/OPG pathway: genetic association with history of ischemic stroke in Italian population. Eur Rev Med Pharmacol Sci 2016, 20, 4574-4580. 

Lyons, M.; Smith, C.; Boaden, E.; Brady, M.C.; Brocklehurst, P.; Dickinson, H.; Hamdy, S.; Higham, S.; Langhorne, P.; Lightbody, C,; McCracken, G.; Medina-Lara, A.; Sproson, L.; Walls, A.; Watkins, D.C. Oral care after stroke: Where are we now? Eur Stroke J. 2018 Dec;3(4):347-354. doi: 10.1177/2396987318775206.

Cieplik, F.; Wiedenhofer, A.M.; Pietsch, V.; Hiller, K.A.; Hiergeist, A.; Wagner, A.; Baldaranov, D.; Linker, R.A.; Jantsch, J.; Buchalla, W.; Schlachetzki, F.; Gessner, A. Oral Health, Oral Microbiota, and Incidence of Stroke-Associated Pneumonia-A Prospective Observational Study. Front Neurol. 2020;11:528056. doi: 10.3389/fneur.2020.528056.

Reviewer 2 Report

It was great to see an ongoing study on this topic. The association between ischemic stroke and oral health has been reported previously, but there is a paucity of studies in the literature. I have some minor questions/suggestions below.

1-Please, update the references in your introduction. There are several systematic reviews published in the past two years reporting the association between stroke and poor oral health (periodontal disease, tooth loss). These reviews also support your argument that more clinical studies are needed in the field.

2-What was the inclusion criteria regarding the number of teeth?

3-What was the tooth-brushing technique performed/taught to the participants?

4-Have the authors considered the inclusion of a control group without cerebrovascular accidents? Or presenting cognitive decline? It would be great to compare the progression between these groups.

5-Please state the limitations of the study in the Discussion section.

Author Response

Concern of the Reviewer: It was great to see an ongoing study on this topic. The association between ischemic stroke and oral health has been reported previously, but there is a paucity of studies in the literature. I have some minor questions/suggestions below.

Please, update the references in your introduction. There are several systematic reviews published in the past two years reporting the association between stroke and poor oral health (periodontal disease, tooth loss). These reviews also support your argument that more clinical studies are needed in the field.

Authors’ response: Thank you for this suggestion. We updated in introduction several systematic reviews reporting the association between stroke and poor oral health.

“In recent meta-analyses authors compared oral health between stroke patients and controls (healthy population). Results indicated, that stroke patients had poorer oral health and worse periodontal status [8-10] as well as less frequent dental attendance behaviour than controls [11,12]. Therefore, further research concerning oral health problems after stroke should be conducted and effective management strategies need to be devised and implemented. It must be emphasized that only a few studies have evaluated oral health status according to stroke patients' functional level [13,14]. Tools are needed to measure progress in achieving daily activity of rehabilitated stroke patients. [15]. The quality of rehabilitation in stroke patients can be measured according to the level of independence achieved in daily activities. This goal seems to be of paramount importance in the rehabilitation of this group of patients. An adequate and rigorous assessment is vital for the rehabilitation physician (physiatrist) to establish proper physiotherapy and physical recovery outcomes, monitor progress in patients’ rehabilitation, and prescribe an adequate medicine treatment [16] with particular attention to the polypharmacy and its possible consequence [17].”

Lafon, A.; Pereira, B.; Dufour, T.; Rigouby, V.; Giroud, M.; Béjot, Y.; Tubert-Jeannin, S. Peridontal disease and stroke: a met-analysis of cohort studies. Eur J Neurol. 2014, 21, 1155-61, e66-7. doi: 10.1111/ene.12415.

Fagundes, N.C.F.; Almeida, A.P.C.P.S.C.; Vilhena, K.F.B.; Magno, M.B.; Maia, L.C.; Lima, R.R. Periodontitis as a risk factor for stroke: a systematic review and meta-analysis. Vasc Health Risk Manag. 2019, 15, 519-532. doi: 10.2147/VHRM.S204097.

Zeng, L.N.; Rao, W.W.; Luo, S.H.; Zhang, Q.E.; Hall, B.J.; Ungvari, G.S.; Chen, L.G.; Xiang, Y.T. Oral health in patients with stroke: a meta-analysis of comparative studies. Top Stroke Rehabil. 2020, 27, 75‐80, doi:10.1080/10749357.2019.1656413.

Dai, R.; Lam, O.L.; Lo, E.C.; Li, L.S.; Wen, Y.; McGrath, C. A systematic review and meta-analysis of clinical, microbiological, and behavioural aspects of oral health among patients with stroke. J Dent. 2015, 43, 171-80, doi: 10.1016/j.jdent.2014.06.005.

Ajwani, S.; Jayanti, S.; Burkolter, N.; Anderson, C.; Bhole, S.; Itaoui, R.; George, A. Integrated oral health care for stroke patients - a scoping review. J Clin Nurs. 2017, 26, 891-901, doi: 10.1111/jocn.13520.

Roman, N.; Miclaus, R.; Repanovici, A.; Nicolau, C. Equal Opportunities for Stroke Survivors’ Rehabilitation: A Study on the Validity of the Upper Extremity Fugl-Meyer Assessment Scale Translated and Adapted into Romanian. Medicina 2020, 56, 409, doi:10.3390/medicina56080409.

Vetrano, D.L.; Villani, E.R.; Grande, G.; Giovannini, S.; Cipriani, M.C.; Gravina, E.M.; Bernabei, R.; Onder, G. Association of Polypharmacy With 1-Year Trajectories of Cognitive and Physical Function in Nursing Home Residents: Results From a Multicenter European Study. J Am Med Dir Assoc 2018, 19, 710-713, doi: 10.1016/j.jamda.2018.04.008.

“Needless to say, intensive rehabilitation is necessary as soon as possible after stroke incident to better improve motor, cognitive, and daily functioning of those group of patients. Moreover, rehabilitation presents a positive effect on some predictive elements such as oxidative process [20,21]. However, medical professionals should consider that several factors can impact on recovery in hospitalized patients [22-24]. Despite establishing a relationship between oral health problems and several medical conditions, the need for cooperation between medical and dental professionals is little known regarding the impact of oral health problems on patients' advances in rehabilitation treatment after stroke.

It must be also emphasized that knowledge concerning proper oral care in stroke patients that stay, e.g, at the rehabilitation centers and nursing home settings [25] is lacking. Some future research should focus on potential associations between diseases of oral cavity and neglected oral hygiene [26]. Thus, more clinical studies are needed in the field to support medical professionals.”

Giovannini, S.; Onder, G.; Leeuwenburgh, C.; Carter, Ch.; Marzetti, E.; Russo, A.; Capoluongo, E.; Pahor, M.; Bernabei, R.; Landi, F. Myeloperoxidase levels and mortality in frail community-living elderly individuals. J Gerontol A Biol Sci Med Sci 2010, 65, 369-76, doi: 10.1093/gerona/glp183.

Giovannini, S.; Onder, G.; Lattanzio, F.; Bustacchini, S.; Di Stefano, G.; Moresi, R.; Russo, A.; Bernabei, R.; Landi, F. Selenium Concentrations and Mortality among Community-Dwelling Older Adults: Results from IlSIRENTE Study. J Nutr Health Aging 2018, 22, 608-612. doi: 10.1007/s12603-018-1021-9.

Padua, I.; Imbimbo, I.; Aprile, I.; Loreti, C.; Germanotta, M.; Coraci, D.; Piccinini, G.; Pazzaglia, C.; Santilli, C.; Cruciani, A.; Carrozza, M.C.; FDG Robotic Rehabilitation Group†. Cognitive reserve as a useful variable to address robotic or conventional upper limb rehabilitation treatment after stroke: a multicentre study of the Fondazione Don Carlo Gnocchi. Eur J Neurol 2020, 27, 392-398, doi: 10.1111/ene.14090.

Vetrano, D.L.; Collamati, A.; Magnavita, N.; Sowa, A.; Topinkova, E.; Finne-Soveri, H.; van der Roest, H.G.; Tobiasz-Adamczyk, B.; Giovannini, S.; Ricciardi, W.; Bernabei, R.; Onder, G.; Poscia, A. Health determinants and survival in nursing home residents in Europe: Results from the SHELTER study. Maturitas. 2018, 107, 19-25. doi: 10.1016/j.maturitas.2017.09.014.

Biscetti, F.; Giovannini, S.; Straface, G.; Bertucci, F.; Angelini, F.; Porreca, C.; Landolfi, R.; Flex, A.  RANK/RANKL/OPG pathway: genetic association with history of ischemic stroke in Italian population. Eur Rev Med Pharmacol Sci 2016, 20, 4574-4580. 

Lyons, M.; Smith, C.; Boaden, E.; Brady, M.C.; Brocklehurst, P.; Dickinson, H.; Hamdy, S.; Higham, S.; Langhorne, P.; Lightbody, C,; McCracken, G.; Medina-Lara, A.; Sproson, L.; Walls, A.; Watkins, D.C. Oral care after stroke: Where are we now? Eur Stroke J. 2018 Dec;3(4):347-354. doi: 10.1177/2396987318775206.

Cieplik, F.; Wiedenhofer, A.M.; Pietsch, V.; Hiller, K.A.; Hiergeist, A.; Wagner, A.; Baldaranov, D.; Linker, R.A.; Jantsch, J.; Buchalla, W.; Schlachetzki, F.; Gessner, A. Oral Health, Oral Microbiota, and Incidence of Stroke-Associated Pneumonia-A Prospective Observational Study. Front Neurol. 2020;11:528056. doi: 10.3389/fneur.2020.528056.

Concern of the Reviewer: What was the inclusion criteria regarding the number of teeth?

Authors’ response: We have added following information:

“In the present research, patients even with single teeth were taken into consideration bearing in mind the fact that regardless of the number of teeth present in the oral cavity, it is necessary to take care of oral hygiene properly. On the other hand, high DMFT index shows previous dental problems, i.e., dental caries and its complications which might have deleterious effect on the organism and affect general health of the individual.”

Concern of the Reviewer: What was the tooth-brushing technique performed/taught to the participants?

Authors’ response: We have added information tooth-brushing technique performed and taught to the patients.

“Indeed, during the dental examination at the Center the dentists had a unique opportunity to instruct the patients about oral hygiene. They recommended systematic, twice a day, use of soft toothbrush and normal toothpaste, after breakfast and in the evening, and rinsing of the mouth after lunch as it was recommended in the study of Murray and Scholten [19]. Mostly the modified Bass method was applied to the individuals as it is easy to learn, most effective and most widely accepted technique of toothbrushing [52]. This method was also used by other dentists in stroke patients [53]. Needless to say that proper hygienic requirements concerning dentures were also given. In numerous cases there was a need to for assistance of nurse or caregiver during the procedure, generally upon admission to the center, since the patients were not self-dependent due to problems with cognition, arm weakness or aphasia. Interestingly, literature data points to the lack of staff training in oral hygiene techniques and oral assessments [54]. Therefore, it was indicated that training should be provided to nurses, nursing students, and health care assistants by qualified professionals such as dentists.”

Kumar, G.; Jalaluddin, M.; Singh, D.K. Tooth Brush and Brushing Technique. Journal of Advances in Medicine 2013;2,1:65-76.

Chen, H.J.;  Chen, J.L.;  Chen, C.Y.; Lee, M.;  Chang, W.H.;  Huang, T.T. Effect of an Oral Health Programme on Oral Health, Oral Intake, and Nutrition in Patients with Stroke and Dysphagia in Taiwan: A Randomised Controlled Trial. Int J Environ Res Public Health. 2019;16(12):2228. doi: 10.3390/ijerph16122228.

Woon, C. Improving oral hygiene for stroke patients. Australasian Journal of Neuroscience 2017;27,1:11-13.

Concern of the Reviewer: Have the authors considered the inclusion of a control group without cerebrovascular accidents? Or presenting cognitive decline? It would be great to compare the progression between these groups.

Authors’ response: Thank you very much for your suggestion. It would be very interesting to compare both groups, i.e., patients after stroke and those without cerebrovascular accidents. Therefore, it would be a subject of forthcoming study. We will design the research to make observations and to compare the progression between these groups during the same time of observations.

Concern of the Reviewer: Please state the limitations of the study in the Discussion section.

Authors’ response: We have added information concerning the limitations of the study in the Discussion section.

„Indeed, the present research has some limitations that require discussion. Firstly, we have not included individuals that have not suffered cerebrovascular accidents or patients after stroke that were not undergoing intensive neurorehabilitation therapy. It would be interesting to make observations and to compare the progression between these groups during the same time of observations. Secondly, radiographs of the patients’ teeth for diagnosis of interproximal caries were not taken, consequently, some carious cavities might be underestimated.”
